# Perspectives in Genome-Editing Techniques for Livestock

**DOI:** 10.3390/ani13162580

**Published:** 2023-08-10

**Authors:** Julia Popova, Victoria Bets, Elena Kozhevnikova

**Affiliations:** 1Laboratory of Bioengineering, Novosibirsk State Agrarian University, 630039 Novosibirsk, Russia; popova@mcb.nsc.ru (J.P.); vish22@yandex.ru (V.B.); 2Center of Technological Excellence, Novosibirsk State Technical University, 630073 Novosibirsk, Russia; 3Laboratory of Experimental Models of Cognitive and Emotional Disorders, Scientific-Research Institute of Neurosciences and Medicine, 630117 Novosibirsk, Russia

**Keywords:** genome editing, livestock, CRISPR/Cas9

## Abstract

**Simple Summary:**

Genetically modified farm animals have been actively used in a number of applications from testing gene functions to increasing production and economic value of livestock. To date, industrial genome engineering in production animals has been successfully applied to increase economically significant traits, reduce the health risk of animal products and express recombinant proteins for the needs of the pharmaceutical industry. Future applications of transgenic animals extend to producing xenografts for medical uses as well as many other promising areas. Here we review the perspectives of genome engineering in livestock from the technical point of view.

**Abstract:**

Genome editing of farm animals has undeniable practical applications. It helps to improve production traits, enhances the economic value of livestock, and increases disease resistance. Gene-modified animals are also used for biomedical research and drug production and demonstrate the potential to be used as xenograft donors for humans. The recent discovery of site-specific nucleases that allow precision genome editing of a single-cell embryo (or embryonic stem cells) and the development of new embryological delivery manipulations have revolutionized the transgenesis field. These relatively new approaches have already proven to be efficient and reliable for genome engineering and have wide potential for use in agriculture. A number of advanced methodologies have been tested in laboratory models and might be considered for application in livestock animals. At the same time, these methods must meet the requirements of safety, efficiency and availability of their application for a wide range of farm animals. This review aims at covering a brief history of livestock animal genome engineering and outlines possible future directions to design optimal and cost-effective tools for transgenesis in farm species.

## 1. Introduction

More than 10 thousand years have passed since the domestication of various animal species by humans. Since then, much has changed in approaches to animal breeding. The modern agricultural industry is developing at an intensive pace and setting new challenges like the safety of breeding technologies and reducing the impact on the environment [1,2]. The traditional approach to livestock and domestic animal breeding relies on the positive and negative selection of the desired characteristics and usually takes generations to select and stabilize genetic traits in a population. In contrast, the relatively recent technologies of genome editing offer a one-step generation of an animal with predefined genetic and phenotypic features based on our knowledge of gene functions [3]. The term “GMO” (genetically modified organism) is used to refer to an organism whose genome has been deliberately altered in any way, including alterations that can occur naturally [4]. To date, the science of creating transgenic animals is one of the fastest-growing biotechnology fields, as it promises many advantages in a short period of time. However, even though genome engineering is speedy and often offers precise control over the traits, it raises strong safety concerns regarding the expansion and use of genetic modifications in contrast to traditional breeding, which is envisioned as safe [5,6].

In the past decades, the rapid expansion of our knowledge regarding the molecular and genetic mechanisms underlying cellular events allowed precise genetic engineering of a given process. Thus, genetic modification of laboratory rodents was commonly implemented more than 40 years back, which soon led to the first attempts to transfer this technology to farm animals [7,8]. As a result, applied genetic engineering in breeding and biotechnology also underwent rapid development [9]. Despite all challenges, genetically modified (GM) farm animals have been produced, including cattle, sheep, goats, pigs, rabbits, chickens and fish [2].

Research in the field of gene engineering of farm animals has mainly focused on improving the efficiency of food production and animal health and welfare. Attempts are also made at reducing the impact of livestock products on human health and the environment [10,11]. In medical research, gene editing of farm animals can be used in a wide range of applications, from large-scale protein expression to the creation of humanized organs for transplantation (xenografts) [12,13,14]. Some large animals can be successfully used as preclinical models in testing drugs, artificial implants or surgical procedures [15,16]. The major strategy for agricultural GM clones is the maintenance of breeding stocks, and not productive stocks per se. Such breeding animals allow the creation of highly effective healthy herds by increasing the number of breeders with the desired traits. Subsequently, these animals are used for conventional breeding, and the offspring obtained from them can be used in production [17]. 

Striking examples of pioneering experiments in obtaining transgenic farm animals are the knock-out pig and ferret models for CFTR (Cystic Fibrosis Transmembrane Conductance Regulator). These larger CFTR animal models recapitulate many phenotypic characteristics of the human disease, which are clinically much more relevant than those found in the corresponding knock-out mice [18]. Increasing weight gain in beef cattle and pigs is the top priority for the industry. One example of the superior efficiency of transgenic animals in production is GM gilts that produce 70% more milk than the control non-transgenic littermates. The offspring of these gilts will grow 500 g larger in 21 days of lactation [19]. As another example, Dr. Alison Van Eenennaam’s team proposes a genome-engineered solution to produce hornless cow breeds with superior muscle mass [20,21]. An alternative approach to increase productivity is to manipulate the reproductive performance of livestock. Thus, one study involved the introduction of a point mutation in the *GDF9* (growth differentiation factor 9) gene, which had a large influence on both the rate of ovulation and the size of the litter in goats [22]. Likewise, another study showed that a mutation in the *BMPR-1B* (*FecB*) gene in sheep leads to an increased rate of ovulation and, consequently, to an increase in litter size [23].

Not long ago, researchers introduced a mouse gene that regulates body temperature into a pig. This improvement allowed the pigs to maintain a physiologically optimal temperature in cold weather by burning fat, which resulted in 24% less adipose tissue than in normal pigs [24]. This genomic change offers the production of pork with a low fat content, which may be considered beneficial to humans. At the same time, such transgenes are less demanding in terms of temperature maintenance in a facility, affording refined economic efficiency of such GM animals [24]. Other essential needs in livestock maintenance are successfully addressed by means of genome engineering. For instance, transgenic goats bearing knock-out mutations in two genes that inhibit hair and muscle growth allow breeders to produce more cashmere and meat [25].

Targeted mutagenesis has also been applied to the genes encoding ovalbumin and ovomucoid, common egg allergens in chickens, using the clusters of regularly interspaced short palindromic repeats (CRISPR) method. The researchers hope to apply this technique in producing hypoallergenic eggs in the future [26].

Sheep and pigs are actively used for genome editing in order to recapitulate key features of human diseases. Williams et al. demonstrate a sheep model that reproduces human hypophosphatasia, a rare metabolic bone disease, using CRISPR/Cas9 (CRISPR-associated nuclease 9) as a molecular tool [27]. Likewise, large animals with the *NUP155* knock-out mutation are useful models in heart tissue physiology research [28,29,30]. Fan et al. reported the creation of *IFNAR* knock-out sheep using CRISPR/Cas9 in combination with somatic cell nuclear transfer (SCNT) to produce a large animal model with high susceptibility to Zika virus [31].

It is also worth mentioning a study that created a model with impaired pancreatic development in sheep. The group of scientists applied CRISPR/Cas9 to *PDX1* (pancreatic and duodenal homeobox protein 1), which is essential to pancreatic growth [32,33]. CRISPR/Cas9 combined with a direct oocyte microinjection was used to disrupt *PDX1*, resulting in homozygous mutant fetuses that were pancreas-free. The results of these promising efforts highlight the potential of gene-edited sheep as hosts for human xenograft growth via blastocyst complementation [32,33].

The above examples provide evidence that the prospects for using transgenic animals in the agricultural industry are good. However, such technologies will not reach mass production in the near future. The biosafety of products derived from GM animals and the high costs of obtaining and certifying transgenes are key restraints to the introduction of livestock biotechnologies into agriculture.

## 2. Precision Genome-Editing Tools Successfully Used in Livestock

### 2.1. Zinc Finger Nucleases (ZFNs)

Zinc finger nucleases (ZFNs) are engineered nucleases containing a zinc finger protein domain as a DNA binding component and a DNA cleavage domain fused together to form an artificial restriction enzyme [34]. Zinc finger domains are designed to bind specific nucleotide combinations within DNA. This allows ZFNs to target unique sequences within complex genomes. Each zinc finger domain binds three-nucleotide sequences in a sequence-specific manner. Combining several “zinc finger” domains allows ZFNs to recognize virtually any DNA sequence. The most commonly used nuclease is the catalytic domain of the restriction enzyme *FokI*, which consists of two subunits, so nucleases work in pairs. The recognition sites are chosen so that the distances between them are sufficient to dimerize *FokI* domains and form a catalytically active structure [34].

The main allergen found in goat and cow milk is β-lactoglobulin, which is not present in human milk. Genetic engineers had a long-term objective of producing milk from animal sources that is depleted of β-lactoglobulin and contains a biologically active human protein. The task was to “humanize” goat milk by integrating human milk protein genes, lactoferrin or lactalbumin, into the endogenous gene region. ZFN technology was used to solve this long-standing issue by creating gene-edited cattle in the β-lactoglobulin locus [35].

Other successful examples of applied ZFN technology in farm animals involve pigs and cows. For example, *GGTA1* (α-1,3-galactosyltransferase) was successfully knocked out in pigs—the first step in creating transgenic swine donors [36]. Likewise, Liu and co-authors used a ZFN to produce transgenic cows carrying the human lysozyme gene in the bovine β-casein locus [37]. Such animals are resistant to mastitis since the milk secreted by transgenic cows has the ability to inactivate *Staphylococcus aureus* [2].

Injection of ZFN mRNA or DNA into zygotes has been successfully used to edit the genome in rabbits and rodents [38,39]. ZFN in-embryo editing helped to efficiently achieve interspecies allele introgression in one generation in pigs [40]. This method has several disadvantages, including the possibility of off-target cleavage by zinc finger domains. The method is also labor-intensive, as it requires creating a ZFN protein structure for every DNA sequence. Thus, the “zinc finger” system has not been widely adopted [10].

### 2.2. Transcription Activator-like Effector Nucleases (TALENs)

Another relevant and widely used method of genome editing of farm animals today is based on chimeric nucleases called transcription activator-like effector nucleases (TALENs) [41,42]. The TAL protein domains are responsible for the recognition of specific nucleotides and can be combined to create a sequence-specific binding subunit, which in turn is fused to a DNA cleavage domain of a *FokI* endonuclease. Such proteins usually contain a DNA-binding domain consisting of 33–35 consecutive amino acid repeats that specifically bind to the host’s genomic DNA [43]. TAL-domain prototype proteins come from *Xanthomonas* bacteria where they facilitate plant infection [34]. In 2011, this approach was awarded a “Method of the Year” award in *Nature Methods* due to the wide range of possible applications in the fields of basic and applied science, from functional genomics to developmental biology and agricultural biotechnology [44]. 

The development of TALEN technology facilitated the knock-in strategy, the essence of which is the integration of target genes into specific regions of the genome. This strategy has been used to replace *BSA* (bovine serum albumin) with two *hSA* (human serum albumin) minigenes to specifically express them in the liver and mammary gland in cattle [45]. The knock-in strategy appeared as a promising way to create cows producing recombinant therapeutic proteins in milk [41].

In 2015, TALEN technology was used to perform the genetic editing of bulls [20]. A *POLLED* mutation has been introduced, which is sometimes found in the wild and is associated with the lack of horns. For agricultural breeds, this mutation could be a valuable asset, as horns make animals more dangerous to personnel. Now, bulls have their horns removed, but it was considered more cost-effective and convenient to create a hornless bull breed. Researchers chose TALEN technology to edit genomes in the connective tissue cells of an adult bull. Then, the nuclei were collected and transferred to the eggs of cows by reproductive cloning [20]. Amy Young and colleagues tested the genome of the edited bulls and evaluated the efficiency of the new mutation’s transmission to the next generation [21]. As a result, the edited bull produced six offspring, all hornless. All six calves were heterozygotes, carrying one wild-type and one dominant mutant allele. DNA testing showed that no traces of TALEN off-target activity, i.e., no unplanned mutations, were found. The offspring showed no physiological abnormalities, or any health disorders, with the exception of one bull whose testicle did not descend. The remaining four were tested according to all veterinary standards and were recognized as potential sires [21]. It was also shown that GM offspring had no effect on the mothers. Microchimerism in mammals is common when fetal cells colonize the red bone marrow of the mother. Scientists could not detect any mutant genes in the cows’ blood, either during or after pregnancy. They confirmed that genetic modification did not pass from fetuses to mothers in any way [46].

However, despite some advances in the production of GM animals, TALEN technology has a number of limitations. Its high cost, combined with the difficulty of constructing recombinant vectors, makes it a less attractive option than traditional knock-out technologies.

### 2.3. Clustered Regularly Interspaced Palindromic Repeats (CRISPR)

New opportunities have appeared following the introduction of the new genome-editing system, which revolutionized the production of GM animals. Le Cong described the first use of the CRISPR/Cas9 system and brought genetic engineering to a new level [47]. 

The mechanism that bacteria use to defend themselves against their pathogenic viruses (bacteriophages) inspired the development of this system [48]. The Cas9 protein makes a double-strand break around the protospacer adjacent motif (PAM), which contains a conserved NGG sequence. Cas9 navigates to the PAM using guide RNA complementary to a 19-nucleotide sequence upstream of the PAM [49]. The non-homologous end-joining repair machinery introduces small deletions or inserts at the site of the break, thereby creating possible mutations. Simultaneous addition of constructs that contain homology arms around the break results in homologous recombination. This facilitates the insertion of the desired fragment at a specific location in the genome. The CRISPR/Cas9 technology has the advantage of allowing multiple genetic constructs to be introduced into cells simultaneously, each targeting a different part of the genome.

CRISPR/Cas9 technology for animal transgenesis finds application in conjunction with microinjection and SCNT techniques. In several reports, it was described that primary goat fibroblasts could be successfully modified at 80–90% efficiency [29,50,51]. Fibroblasts with a diallel knock-out of the myostatin gene were used for SCNT, resulting in viable transgenic offspring [52]. One of the proposed ways to eliminate genetic mosaicism is the direct injection of the CRISPR/Cas9 system components into metaphase II oocytes or early zygotes. Alternatively, electroporation of zygotes with the Cas9-RNP completely eliminates mosaicism [53]. In sheep and cattle, injection of CRISPR/Cas9 into zygotes reduces mosaicism more effectively than injection of metaphase II oocytes [2,54]. At the moment, fetal fibroblast-derived genetic modifications have been created for almost all types of farm animals, including cattle, pigs and goats, using the CRISPR/Cas9 system [29,52,55,56].

The CRISPR/Cas9 system is widely used in many areas of livestock transgenesis. For instance, this system was successfully applied to a long-standing problem of sex selection in farm animals. Previously, various methods have been used to approach this issue: non-radioactive hybridization, fluorescence in situ hybridization, sex chromosome-based PCR and labeled Y-chromosome-specific probes [2]. By knocking in the *eGFP* in the Y chromosome of an embryonic bovine fibroblast (bovine fetal fibroblast (BFF)) system and subsequently transferring the resulting Y-Chr-eGFP construct using the SCNT method, it was possible to determine XY embryos by simply tracing a color label [57].

One of the first genes to be targeted using this technique was *MSTN*, which regulates muscle mass. In 2014–2015, researchers were able to create mutant sheep with higher body weight and muscle mass than their wild-type counterparts by editing this gene using the CRISPR/Cas9 system [58,59]. These initial experiments paved the way for further application of genome editing in small ruminants.

Subsequent studies have demonstrated the versatility of the CRISPR/Cas9 system for multiplex gene editing in sheep. For example, efficient editing of three genes—*MSTN*, *ASIP* (agouti-signaling protein) and *BCO2* (β-carotene oxygenase 2)—was achieved using this method [60,61]. Importantly, these studies showed minimal off-target effects. This could be particularly valuable for generating new lines of farm animals with multiple desired traits, as many economically important features are controlled by multiple loci.

In addition to gene knock-outs, the CRISPR/Cas9 system can also be used to generate animals with specific point mutations. For instance, Zhou et al. reported a high efficiency of single nucleotide substitution in the *SOCS2* gene, which controls body weight, size and milk production in sheep [62]. These findings highlight the tremendous potential of the CRISPR/Cas9 system for creating GM farm animals with improved productivity and economic value.

Genetic editing methods have been mainly used in mammalian species of farm animals, while birds, which are equally important for agriculture and also serve as model organisms in biology, have not yet been a focus of genetic manipulation efforts. However, the chicken is one of the most widely farmed bird species and has been instrumental in several scientific breakthroughs. For instance, the first cholera vaccine developed by Louis Pasteur was tested in 1878 on domestic chickens. Additionally, the chicken was the first non-mammalian species to have its genome sequenced, and comparative analysis of the chicken genome has aided in the discovery of new genes and their functions in both animals and humans [63].

Therefore, using the CRISPR/Cas9 system to produce GM chickens with desirable traits could be highly beneficial for agricultural and industrial applications. However, adapting the genetic editing tools that are commonly used in mammals to birds poses significant challenges due to differences in development and physiology. In particular, accessing and manipulating the fertilized oocyte nucleus in birds presents technical difficulties [64,65].

Despite multiple challenges, scientists have successfully implemented the CRISPR/Cas9 system in avian species. For instance, Koslová et al. were able to produce chickens that are resistant to avian leukosis virus (ALV) [66]. Another notable example is the generation of *MSTN* knock-out chickens, wherein nickase D10A-Cas9 (Cas9n), a mutant Cas9 protein, was employed to minimize any non-specific double-strand breaks and off-target effects [67]. In mammals, mutations in the *MSTN* gene result in a distinct “double” musculature phenotype due to accelerated muscle growth. Similarly, *MSTN* knock-out chickens and quails exhibit significantly larger skeletal muscles [67,68,69,70]. Furthermore, Park et al. were able to create *G0S2* (G0/G1 switch gene 2) gene-edited chickens utilizing the CRISPR/Cas9 system [71]. *G0S2* serves as an inhibitor of adipose triglyceride lipase (ATGL, also known as protein 2 containing a patatin-like phospholipase domain), which catalyzes the first step of lipolysis (hydrolysis of triacylglycerols to diacylglycerols). The resulting *G0S2* knock-out chickens can be utilized as model animals in studies on obesity, providing an alternative to rodents [72].

Recent advances in multiplex gene editing, single nucleotide substitution and the application of CRISPR/Cas9 technology to birds demonstrate its great potential for agricultural and industrial applications, while also providing new animal models for human and animal research.

## 3. Methods of Gene Delivery Successfully Applied in Livestock Biotechnology

The production of GM animals is associated with changes in their genomes at early embryonic stages. This technically complex process consists of three key stages: (1) isolation of zygotes from females, (2) delivery of nucleic acids (NAs) into isolated zygotes and (3) subsequent transfer of embryos to pseudo-pregnant female recipients to further obtain viable offspring. To be successful, these steps require advanced equipment and a series of well-timed procedures performed by experienced and qualified personnel. The delivery of NAs to embryos is carried out predominantly by microinjection. There are other, less commonly used methods of NA delivery, such as electroporation-mediated gene transfer, viral transduction using adeno-, retro- and lentiviral vectors, and liposomal transfection [73,74]. In turn, one of the most advanced strategies for propagating genetically engineered or genome-edited specimens in livestock species seems to be cloning by SCNT [75,76,77,78]. With the application of SCNT-mediated cloning, genetically transformed progeny can be either created with the use of *in vitro* transfected nuclear donors (somatic or stem cells) or multiplied with the use of ex vivo expanded nuclear donor cells derived from existing gene-edited farm animals that have been previously generated by other techniques of gene delivery [79,80,81,82]. A broad spectrum of the standard and more advanced techniques of gene delivery is presented in Figure 1.

### 3.1. Microinjection into the Pronucleus

The pronucleus microinjection method developed in 1980 was the first in genetic modification of animals, and in subsequent decades, it was the most commonly used method [83,84,85]. With this method, a genetic construct is introduced into a fertilized egg via direct mechanical injection into the pronucleus using a micropipette. The construct is randomly integrated into the DNA of a zygote. Since the newly inserted DNA was never essentially part of the host genome, the animals obtained by this method were called transgenic [15]. The pronucleus microinjection method has many practical disadvantages: integration into the genome is random; the maximum size of constructs is limited. Therefore, genes are almost always introduced in their fully spliced isoforms. Thus, the remaining gene regulatory mechanisms were usually rudimentary, and transgenes tended to be silenced over several generations.

In farm animals, however, the issue is complicated by the biology of reproduction. In mice, the efficiency of transgene introduction was about 5–10%. With a gestation period of 3 weeks and a litter size of 6–10 pups (depending on the animal line), five female recipients are likely to produce several transgenic mice in just a few weeks. In farm animals, however, for unclear reasons, the efficiency of transgene insertion was closer to 3%, while only 18% of the blastocysts were able to produce live calves, which reduced the efficiency to a fraction of a percent. With one calf per female, this meant hundreds of attempts were required to successfully create a transgenic animal [86,87,88]. The advance of a relatively routine SCNT about 20 years ago partially solved this problem, because cells could be tested for correct insertion prior to implantation in recipient mothers. However, other limitations related to the use of transgenes remained. Despite all complications mentioned above, transgenic sheep, pigs, goats and cattle have been created via pronuclear microinjection, but most of them have low practical utility [7]. Several scientific groups have successfully overcome the problem of low insertion efficiency by using lentiviral transgenic vectors [89].

The groundbreaking technology that offered control over gene editing is the invention of precise genome targeting. It allows utilizing native promoter elements and natural regulation of splicing to properly regulate genes, therefore eliminating variegation and repression of genes associated with random insertions. The technology essentially operates by molecular scissors known as ZFN, TALEN and CRISPR/Cas9 to introduce genetic changes into a precisely controlled genomic environment [90]. The probability of the off-target events was shown to be very low when combining the CRISPR/Cas9 system with microinjections into zygotes [91]. The Cas9-assisted genome editing via pronuclear microinjection method was successfully implemented in mice, rats, monkeys and other animals [91,92,93]. 

The microinjection method itself, which was originally created to obtain transgenic mice, was the first technique successfully applied in the production of transgenic farm animals [83,85,94]. In detail, the method involves introducing a solution of gene constructs in the form of linear DNA molecules into the male pronucleus of a zygote as it is larger in size and therefore easier to visualize than the female one, or the introduction of RNA into the cytoplasm [95,96,97,98]. The microinjected zygotes are further cultivated under *in vitro* conditions to the stage of preimplantation embryos and then transferred to the recipient female’s reproductive system. The efficiency of microinjection depends, among other things, on the purity of the solution, concentration and quality of DNA and RNA, size of the injected construct, operator’s skills and stage of embryo development. Due to the low efficiency of this procedure, around 10% in mice, 2–3% in pigs, 4% in rabbits and less than 1% in cattle, researchers are looking for modifications to increase the success rate of microinjections and further transgenesis [99,100,101]. 

The first modification was the simultaneous injection of the transgene into both pronuclei. The increased efficiency of transgene integration was accompanied by abundant injuries during the process, which resulted in high embryonic mortality [102]. Subsequently, microinjection directly into the cytoplasm of the embryo was attempted. With this approach, the problem of the high lethality of injections was eliminated. However, the efficiency of the procedure substantially decreased [103]. Researchers have overcome this problem by introducing genetic material into the cytoplasm in the form of RNA. In this method, RNA stability has been shown to be a critical point. The third version of the DNA microinjection protocol was the delivery of the transgene into the cell nuclei of a two-blastomere embryo. This idea was aimed at increasing the survival rate of embryos. However, due to the higher chance of mosaicism, it is rarely used [104,105]. In order to reduce the occurrence of mosaicism, Tanihara and colleagues conducted a study to find the best time point for injecting the CRISPR/Cas9 system components. They showed that the optimal time for microinjection of the CRISPR/Cas9 RNA–protein mixtures into zygotes is 6 h after initiation of the *in vitro* fertilization (IVF) procedure [105]. At this point, it is possible to promote genome-editing events with minimal impact on the viability of the embryo. Recent research showed that the microinjection of CRISPR/Cas9 components into porcine germinal vesicle oocytes is effective at reducing the risk of mosaicism [106]. After microinjection, oocytes matured *in vitro* and were subjected to parthenogenetic activation or IVF. This strategy helped to produce about 83% of the resulting mutant embryos as non-mosaic [106].

Although the microinjection method is widely used to create edited animal models, it requires expensive micromanipulation equipment and skilled personnel. In addition, it is time-consuming, so the number of microinjections per attempt is limited. As a successful microinjection example, Hai et al. obtained pigs with the knock-out of *vWF*, a gene responsible for the von Willebrand disease in humans [107]. Knock-out of *vWF* was proven in 63% of the resulting offspring (10 out of 16), including six piglets in the diallel state. The overall efficiency of transgenesis was 13.2%. Whitworth et al. generated *CD163* and *CD1D* knock-out transgenic pigs using both SCNT and cytoplasmic injection into zygotes *in vitro* [108]. Targeting efficiency in both cases was 100%.

### 3.2. Retro- and Lentiviral Vectors

Retroviral and lentiviral vectors are effective tools for introducing genes into animal embryonic lines. They can stably integrate into the host genomes, and the relatively small sizes of their genomes enable efficient manipulation *in vitro* [109]. The difference between the two types of vectors is that retroviral vectors can only integrate into actively dividing cells, while lentiviruses can replicate in both dividing and non-dividing cells. The advantage of using retroviral vectors is the possibility of achieving a transgenesis efficiency of 100%; the disadvantage is their limited capacity of the insert carriage (no more than 8000 base pairs) [109]. Thus, intron sequences, like other distal or proximal regulatory elements, play an important role in efficient gene expression and might not be included in viral vectors due to size limitations [110]. Success in the production of transgenic cattle has shown that retroviral vectors can serve as a good alternative for efficient transgenesis in farm animals [109,111]. Although lentiviral vectors have been effectively used to produce transgenic pigs, cows and chickens, their widespread use in livestock is mainly limited to the field of gene therapy [112,113,114]. 

### 3.3. Electroporation of Zygotes

Zygote electroporation is a simplified and streamlined approach to transfection of mammalian zygotes. As it is cheaper and easier than microinjection, it is suitable for introducing indel mutations, large deletions and small insertions [115,116]. The electroporation method allows faster gene editing in a large number of oocytes or zygotes at a time, in contrast to microinjection. The combination of mass production of IVF embryos with the gene editing by electroporation of Cas9 protein (GEEP) method compensates for poor IVF results with high editing speed. Therefore, it became possible to transfer up to 200 embryos per female recipient and, finally, to obtain live offspring with targeted gene modifications [117].

There are a number of studies investigating the variables used in the electroporation of mouse and rat zygotes. These authors reviewed the electroporation conditions, timing and success rates of the procedure in mice and rats, in addition to data on livestock zygotes, particularly pigs and cattle [118]. The administration of editing reagents during or shortly after fertilization helps to reduce the level of mosaicism. Also, the introduction of nuclease proteins rather than nuclease-coding mRNAs significantly improved the efficiency of transgenesis. Mosaicism is especially problematic in large livestock species with long reproductive cycles since it can take years to produce non-mosaic homozygous offspring through crossbreeding. 

A critical step in zygote electroporation is the delivery of large DNA plasmids into the zygote through the *zona pellucida*, and in most cases, short single-stranded DNA (ssDNA) repair templates, usually less than 1 kb long, were used in the electroporation method [118]. The most promising approach to deliver larger donor repair constructs up to 4.9 kb in size along with genome-editing reagents without the use of cytoplasmic injection into zygotes is the use of recombinant adeno-associated viruses in combination with electroporation. However, like other methods, this approach is also associated with a high level of mosaicism. Some protocols use partial dissolution of the *zona pellucida* to facilitate entry of the single guide RNA (sgRNA)/Cas9 complexes (RNPs) into the zygote. However, this can affect the quality and survival of embryos [119,120,121,122,123]. Recent studies have shown that weakening of the *zona pellucida* is not necessary for the successful procedure of gene editing porcine zygotes using custom-designed electroporation protocols [117,124,125]. The NEPA21 electroporator enabled reducing damage to embryos through the use of a three-stage system of electrical pulses. The first pulse (pore-forming) creates micro-holes in the *zona pellucida* and oolemma of the embryos. With the help of the second pulse (transfer), the endonuclease is transferred into the cytoplasm of the embryos. The third transfer pulse with reversed polarity increases the possibility of introducing endonucleases into embryos [126].

While various mutations have been efficiently induced using electroporation-based protocols in mice and porcine embryos, their adaptation to other mammalian species such as cattle has not been fully investigated [53,119,121,124,125,127,128,129]. In a recent study, Miao et al. evaluated the possibility of using electroporation of bovine and porcine zygotes with sgRNA-Cas9 RNPs to generate indel mutations in the *NANOS2* gene [130]. They analyzed different pulsing parameters for bovine and porcine zygotes in order to optimize embryo survival rates. Testing the efficiency of indel mutations in the *NANOS2* gene in porcine embryos at the 2–8-cell stage and in bovine embryos at the 4-cell to blastocyst stage showed that ∼63% (n = 22) contained indel mutations > 300 bp in size in at least one allele of the *NANOS2* gene in cattle and pigs. Both *NANOS2* alleles were edited in 73% of porcine (n = 11) and 82% of bovine (n = 20) embryos [130]. Interestingly, these authors found that bovine zygotes are more sensitive to electroporation than porcine zygotes, which are able to withstand the higher energy of the third pulse. The reason for such a difference is currently unknown, but this result clearly demonstrates the need to adapt electroporation conditions for each species of interest. Also, this study has shown that mutant embryos obtained by electroporation can develop *in vitro* to the blastocyst stage with high efficiency [130]. However, zygotes treated with CRISPR/Cas9 require *in vitro* culturing and further surgical transfer to the reproductive system, which poses risks to embryo survival and increases the cost of the procedure [119,121,131].

### 3.4. Transfer of Nucleus (Cloning, SCNT)

Somatic cell nuclear transfer (SCNT) or cloning is a method in which the nucleus of a somatic cell is transferred into the cytoplasm of an enucleated metaphase II stage ovum to obtain a new animal genetically identical to the somatic cell donor [132]. After the transfer of the somatic nucleus, cytoplasmic factors of an ovum initiate epigenetic reprogramming, which leads to its transformation into the nucleus of a zygote. This method is by far the most commonly used for obtaining transgenic farm animals.

The very first attempts to clone farm animals were performed using embryo splitting (separation). It has been shown that this approach can be used in sheep and cattle to generate genetically identical twins by separate transfer of half-embryos to different female recipients [133,134,135]. However, this separation approach results in the loss of cytoplasmic volume. For this reason, the nuclear transfer (NT) method was developed and first successfully applied in mice, in the early 1980s. By transferring either single-cell stage pronuclei or two-, four-, and eight-cell stage nuclei and the inner cell mass into the cytoplasm of an enucleated egg (cytoplast), viable embryos were produced [134,136]. 

More recently, NT was described in porcine embryos using pronuclear exchange between zygotes, as well as NT between embryos at the two-cell stage [137]. Willadsen was the first to report that blastomere nuclei at the 8- to 16-cell stage are capable of fully developing after NT into enucleated metaphase II oocytes in sheep and cattle [138,139]. After that, other groups were able to use donor nuclei from morulae and inner cell mass embryos at the blastocyst stage or using the re-cloning procedure [140,141,142,143,144,145].

The first successful livestock SCNT was reported by a group led by K. Campbell in 1996, when two cloned lambs were obtained by NT, using a stable differentiated cell line derived from a 9-day-old sheep embryo [146]. One year later, the possibility of obtaining viable offspring by SCNT was demonstrated in sheep using mammary nuclei as donors [146]. This work is world-famous for the birth of Dolly the sheep. The invention of the SCNT technique and the ability to genetically modify fetal or adult somatic donor cells attracted the attention of researchers, and multiple transgenic farm animal models such as sheep, cows, goats and pigs have been created since 1997 [147,148,149,150,151].

Insufficient epigenetic reprogramming is the main cause of SCNT’s inefficiency. It has previously been shown that the survival of female embryos after SCNT was higher than that of male embryos. Abnormal DNA methylation patterns (5-methylcytosine-5 mC and 5-hydroxymethylcytosine-5 hmC) were found in cloned buffalo embryos, which may be due to dosage compensation of the X chromosome, XCI (X-chromosome inactivation) at an early stage of embryonic development [152,153,154]. The *Xist* gene (XCI specific transcription gene) is imprinted on the X chromosome and regulates XCI [155]. The reprogramming ability of donor cells is affected by abnormal expression of the imprinted *Xist* gene and DNA methylation. 

The SCNT method is currently being used to improve the poor fertility of buffalo, which are important livestock in Southeast Asia. However, due to the problem of abnormally low levels of DNA methylation and high expression of the *Xist* gene in transgenic cloned calves, an increase in the proportion of stillborn female cloned buffalo was found compared to naturally breeding female buffaloes [156,157]. All this leads to a decrease in the efficiency of the SCNT procedure. This problem can be solved by suppressing the expression of the *Xist* gene, as was previously done in mice using gene knock-out or siRNA injection; *Xist*-siRNA also led to an increase in the birth rate of healthy cloned male piglets [158,159,160]. It was shown that siRNA-mediated knock-down of the *Xist* gene effectively inhibits gene expression and promotes the development of SCNT-cloned female buffalo embryos [153].

Aside from epigenetic reprogramming of donor nuclei following SCNT, changes in the cytoskeleton also play a role in impaired embryonic development after SCNT. Recent studies have shown that cytoskeletal proteins undergo post-translational modifications, and acetylation of *a*-tubulin at lysine 40 (K40) regulates the cleavage of SCNT embryos [161]. It has been shown that in bovine embryos, the level of *a*-tubulin acetylation varied at different stages of development after fertilization, with the degree of acetylation being much higher in SCNT embryos compared to IVF embryos. This indicates that abnormally high levels of *a*-tubulin acetylation may be a key factor leading to irregular cleavage of SCNT embryos [162,163]. Gao and colleagues attempted to simultaneously correct the abnormal modification of histones and cytoskeletal proteins of SCNT embryos using bovine cells. The restoration of the normal pattern of acetylation affected the development and efficiency of SCNT embryos [164]. 

The potential for GM animals to be produced by SCNT was one of the driving forces behind its development. However, there are several technical obstacles that limit the practical use of this method. First of all, the efficiency of cloning is extremely low in almost all types of farm animals (0.5–1.0%). Second, developmental anomalies are common in extra-embryonic tissues of cloned animals, such as the placenta [165]. Moreover, abnormalities are also observed in cloned animals after birth (obesity, immunodeficiency, defects in the respiratory system), and early death is also a frequent event, although these phenotypes are not transmitted to offspring [165,166,167,168,169].

GM research in farm animals has widely focused on biomedical applications, including xenotransplantation, production of pharmaceutical proteins and animal models of human diseases [170,171,172]. SCNT currently allows the production of gene knock-out and gene knock-in in farm animals [173,174,175,176,177]. However, the use of SCNT in agriculture has lagged primarily due to concerns about the slow production process and the possible public backlash against the use of livestock containing transgenes in the food industry. 

### 3.5. Sperm-Mediated Gene Transfer (SMGT)

Another alternative to obtain transgenic animals is a method called sperm-mediated gene transfer (SMGT). The need to develop a new strategy arose because all methods known at that time in farm animal transgenesis (microinjection into the pronucleus, NT, lentivirus-based methods) had low efficiency and required manipulating early-stage embryos, not to mention high cost [178]. The very first report that rabbit epididymal sperm was able to spontaneously capture and transfer SV40 DNA into the egg at fertilization, thereby causing genetic transformation, was published in 1971 [179]. In 1989, Lavitrano and colleagues reported that mouse epididymal spermatozoa can take up and transfer exogenous plasmid DNA into the ovum [180]. This method was originally used only in mice, and after successful experiments, it was adapted for use in farm animals, especially pigs and cattle [180,181,182,183,184,185,186]. For instance, in pigs, the ability of spermatozoa to bind and internalize exogenous DNA has been shown to be 90% and 70%, respectively [184].

As previously mentioned, different methods of NA delivery are more effective in mice than farm animals. Lavitrano et al. demonstrated that the SMGT method, when used to create pigs for xenotransplantation (pigs with the *hDAF*, human decay accelerating factor), was more efficient than any other method [187]. In eight experiments, 53 of the 93 pigs created were transgenic (57%) [183,184]. This is in contrast to the efficacy of 0.5–4% claimed for microinjections in pigs [188]. The SGMT method was therefore more than 20 times as effective.

Despite its relative simplicity, the SMGT method is not widely used to obtain transgenic farm animals. This is because it has a few disadvantages such as a very low level of gene insertion and rearrangements of the transgene [187,188]. Several attempts have been reported to increase the ability of sperm to incorporate exogenous DNA. These include SMGT based on electroporation, the use of retroviruses, liposomes (lipofection), restriction enzyme-mediated integration and other methods [189]. The SMGT method performed on sheep and goats made it possible to obtain both transgenic embryos and transgenic fetuses using marker transgenes [190,191,192,193,194].

In cattle, SMGT by electroporation has hardly been applied before, except in a study reporting the creation of a transgenic bovine embryo [195]. Goats are a convenient type of small livestock for transgenesis because these animals have a small body size, a short gestation period, high fecundity and a relatively high protein content in milk [196,197]. Pramod et al. reported for the first time the successful optimization of goat sperm electroporation, resulting in maximum absorption of foreign DNA with minimal adverse effects on vital sperm variables, namely progressive motility, viability, membrane integrity and acrosomal response [193]. SMGT-derived goat spermatozoa were then used in insemination using the IVF method.

Various substances have been used to increase the permeability of sperm membranes to increase the efficiency of SMGT. Sánchez-Villalba and co-authors used streptolysin-O to process bovine sperm, followed by an assessment of the intracytoplasmic sperm injection (ICSI) efficiency [198]. Semen taken from bulls treated with streptolysin-O was evaluated for plasma membrane integrity, acrosomal integrity, DNA damage and exogenous DNA-binding capacity (nick translation). Embryonic development and efficiency of transgenesis with the *eGFP* (enhanced green fluorescent protein) marker were also evaluated [198]. The authors found that treatment of bovine spermatozoa with streptolysin-O prior to oocyte infusion with ICSI significantly increased the rate of *eGFP* expression in embryos and the efficiency of the SMGT technique.

Another method for delivering exogenous DNA to sperm is lipofection [199]. Liposomes are small structures composed of membrane-like lipid layers or bilayers that can actually protect foreign DNA from cleavage by proteases and DNases [185]. Cationic liposomes are capable of spontaneously interacting with DNA molecules, forming lipid–DNA complexes. Under appropriate conditions, exogenous DNA can be transferred into cells by the fusion of the lipid–DNA complex with cell membranes, and a portion of this DNA is localized in the nucleus [200]. Cryopreserved bull semen incubated with DNA-containing liposomes was used to show that the distribution of liposomes is random, even between the head and tail of spermatozoa, but only a small proportion of the spermatozoa attached to liposomes. Another disadvantage of this method is the reduced ability of the host genome to incorporate this DNA [201].

Thus, the use of transformed spermatogonia or spermatozoa to produce transgenic farm animals has not yielded significant results, despite some success in obtaining transgenic pigs, cattle and mice [185,186,189,202]. 

## 4. Transgenic Livestock Successfully Approved for Industrial Use

The cost and complexity of the approval process add a major limitation to bringing transgenic animals into mass production. For this reason, despite the long history of animal genome engineering, only a small number of successful cases reached the market shelves. Here we briefly overview the known examples. The fast-growing AquAdvantage salmon was the first GM food animal to be commercialized [203]. The fast-growing fish that expressed *Pacific salmon* growth hormone *gh1* under the control of the antifreeze protein gene promoter was placed on the market by AquaBounty company. The average size of the transgenic salmon was increased by about 2- to 6-fold as compared to the wild-type fish, reaching a 13-fold increase at maximum [204]. The Food and Drug Administration (FDA) has also approved a GM pig for human food consumption and potential therapeutic applications under the commercial name GalSafe [205]. GalSafe pigs have a targeted mutation in the *GGTA1* (alpha1,3-galactosyltransferase) gene that catalyzes the transfer of galactose to acceptors such as extracellular glycans [206]. This enzyme is absent from humans and poses the main risk in using animal donor organs for xenotransplantation [207]. Targeted disruption of this gene in pigs resulted in no detectable alpha-gal sugar on their cell surfaces [173,174,175]. This mutation is also supposed to reduce allergic reactions in humans with alpha-gal syndrome (AGS) to the corresponding sugar found in red meat. The third example of an approved animal for the livestock industry, namely the genome-edited beef cattle produced by a biotechnology company Recombinetics, Inc. (St. Paul, MN, USA), was announced on 7 March 2022 by the FDA [208]. The slick-haired beef cattle, known as PRLR-SLICK cattle, were obtained by an intentional genomic alteration [209]. The *SLICK* hair locus derived from Senepol cattle substantially increases thermotolerance in lactating cows and confers superior thermoregulatory ability [210]. As a prospective example, a large global animal genetics company, United Kingdom-based Genus plc, in collaboration with the University of Missouri, has announced plans to commercialize another genome-edited food animal. This collaboration is planned to introduce and commercialize pathogenic porcine reproductive and respiratory syndrome (PRRS) virus-resistant pigs with a defective *CD163* (PRRS virus receptor) [211].

Meanwhile, there have been several approvals to use transgenic animals in the production of therapeutic human proteins. The well-known first example of a drug from GM animals is human *ATryn1* (antithrombin-III) expressed by goats in milk [212]. These goats are owned by rEVO Biologics, a Massachusetts-based U.S. company, and were generated by pronuclear microinjection. The company received approval for the use of *ATryn1* goats in the U.S. and European Union. In 2014, GM rabbits producing recombinant human C1 esterase inhibitor (Rhucin) in milk were approved to treat hereditary angioedema [213,214]. Another successful example is GM chickens designed to express lipase A, lysosomal acid type (Kanuma), in their eggs, approved in 2015 as a long-term enzyme replacement therapy for the treatment of patients diagnosed with lysosomal acid lipase deficiency [215]. Such chickens are not used to produce meat products. They are bred only to produce therapeutic enzymes from eggs [216].

The most widely known and recognizable GM animal remains a fluorescent aquarium tropical fish named GloFish. These animals were designed to strongly overexpress multicolor fluorescent proteins in skeletal muscle and are famous for their bright colorful glow in the dark [217]. However, the long-term idea behind this technology was the detection of aquatic pollution and water toxins. Information about GM livestock that have been approved for industrial use is presented in Table 1.

## 5. Gene-Editing Systems and Delivery Methods Not yet Applied in Farm Animals or Applied with Little Success

### 5.1. Blastocyst Injection

One way to obtain targeted mutations in animals is to inject embryonic stem cells into diploid embryos (blastocysts). This method was demonstrated back in the late 1980s and is based on the ability of pluripotent stem cells to self-regenerate and develop into the primary layers of germ cells [218,219]. The sources of pluripotent stem cells are embryonic stem (ES) cells and induced pluripotent stem cells [220,221]. However, ES cells per se are not able to implant and cause pregnancy. Therefore, they need a carrier such as a host embryo [222,223]. The injection of ES cells into blastocysts results in a chimeric animal containing both embryonic donor cells and ES cells. When ES cells are injected into tetraploid blastocysts, tetraploid cells only contribute to the placenta and yolk sac and cannot contribute to the somatic cells of the developing mouse, so the resulting pup develops only from the injected ES cells [222]. However, some reports have shown a small proportion of surviving donor tetraploid cells in fetuses [224,225,226]. In mice, genomic editing by microinjection into blastocysts is a relatively routine procedure because ES cells can be maintained in culture for long periods of time, and cell clones effectively contribute to chimeric mouse lines [227].

Adaptation of ES cell cultivation protocols is not yet successful in livestock. As early as 1990, putative ES cells were obtained in the early stages of embryonic development in domestic animals such as sheep, pigs and cattle. However, only a few passages could be maintained in such cell cultures [228]. More recently, cell lines similar to ES cells have been found in pigs, cattle, sheep, goats, horses and buffaloes [229,230,231,232,233,234,235,236,237,238]. At the same time, they are in a state of low self-renewal activity and cannot provide high-quality material for microinjection into blastocysts [228].

### 5.2. Genome Editing via Oviductal Nucleic Acid Delivery (GONAD)

Practical considerations like the number of animals used, tolerance for errors, ethical issues and cost of producing a GM animal force researchers to refine the process of genetic engineering. For instance, a group of Japanese scientists attempted to modify the method of delivering plasmid and transgenic constructs to mouse embryos [239]. They proved that naked plasmid DNA introduced into the lumen of the mouse oviduct on day 1.5 of gestation can be successfully electroporated into two-cell embryos in the oviduct [239]. In 2015, Takahashi and colleagues published an article that was the first to describe an alternative gene delivery method that eliminates all three major/classical steps in animal transgenesis [240]. It has been shown that CRISPR/Cas9-mediated targeted genome editing can be performed without the ex vivo processing of embryos. This method was called genome editing via oviductal nucleic acid delivery (GONAD) and involved editing the genome by delivering NAs into the oviduct. This method is performed in pregnant females carrying E1.5 (E0 of pregnancy is defined as the day a copulatory plug is detected) embryos (two-cell stage) and includes the following steps: (1) performing a surgical operation, during which a 1–1.5 cm dorsolateral incision is made on the anesthetized female; (2) the oviduct, together with the surrounding structures, is carefully removed through an incision on the back; (3) NA solution is injected into the lumen of the oviduct using a glass capillary micropipette; (4) electroporation is performed using tweezer electrodes; (5) tissues are returned to their normal position, and the incision is sutured (Figure 2) [240]. The in situ gene-edited embryos are subsequently developed to term, and the offspring are genotyped for the mutation of interest. Subsequently, the same group of scientists improved the GONAD procedure (hereinafter referred to as improved GONAD (i-GONAD)), including increased efficiency and reduced mosaicism. i-GONAD has another advantage: unlike other methods of genome editing that require female donors to be sacrificed in order to isolate zygotes, the i-GONAD method does not involve the euthanasia of female donors [49,241].

Using the new i-GONAD method, it was possible to obtain a higher efficiency of editing the genome of laboratory animals. It has been demonstrated that i-GONAD can be used to generate germline-modified G1 progeny with a wide range of genetic changes including large deletions and knock-out mutations. In addition, it has been shown that i-GONAD can be performed using various electroporators without affecting the result [241]. This method was also successfully applied in rats and hamsters, suggesting its potential applicability to other animal species of at least comparable size [242,243]. In a recent study, it was demonstrated that employing i-GONAD-mediated electroporation for the delivery of adeno-associated viruses (AAVs) as donors, along with components of the CRISPR/Cas9 system, facilitated the production of knock-out rats harboring a 3 kb gene cassette [243]. These advantages and the ease of implementation make i-GONAD an attractive alternative to the established methods of gene delivery.

### 5.3. Transplacental Gene Delivery (TPGD)

Transplacental gene delivery (TPGD), a method of genomic modification that is rarely used, involves injecting NAs into the bloodstream or fetal tissue [244,245]. Nakamura and colleagues showed that intravenous injection of plasmid DNA with genome-editing components (CRISPR/Cas9 system) can be used to obtain indel mutations in embryonic cardiomyocytes [246]. This method was recently improved by the same group of researchers to reduce the potential embryotoxicity of gene delivery reagents using hydrodynamic force [247].

It was shown that the efficiency of NA delivery to the embryo depends on the time of DNA injection. Thus, it was shown that at E5.5–E9.5, plasmid DNA is introduced with low efficiency. This is because the placenta is immature, the absorption of incoming substances by the visceral endoderm or the yolk sac predominates and most of the introduced material remains in the visceral endoderm [248,249]. It has been determined that at E9.0 when the fetal heart begins to function and other organs differentiate, the efficiency of NA delivery increases [250]. Embryos injected with NAs at E9.0 accumulated at least 40 times more plasmid DNA than those treated at E12.0 or E15.0, whereas plasmid DNA was not detected in fetuses treated at E3.0 or E6.0 [244]. Starting from E10.5 to E13.5, the introduced plasmid DNA is transferred through the blood–placenta barrier and enters the umbilical cord. The second route is from the decidua to the yolk, while some of the DNA enters the yolk sac and is transferred to the embryo after the establishment of functional placental circulation [245,251]. Ideally, TPGD is recommended during mid-gestation, as some successful gene delivery cases have been achieved when TPGD is performed from E9.5 to E12.5 [245]. The main disadvantage of this method is the low efficiency of NA delivery due to the retention of large molecules in the blood–placenta barrier.

This uncommon but promising approach is still at the stage of development, and it is mainly applied to laboratory animal models [245]. However, the TPGD method can be considered as another possible future direction for fetal gene therapy, as well as a strategy to induce protective immunity using plasmid-encoded viral antigens.

## 6. Conclusions

The development of genome-editing techniques has opened new possibilities for improving the health, productivity and medical use of livestock. These techniques offered new venues to introduce beneficial genetic traits and remove harmful ones, leading to useful animal strains and more sustainable agriculture. There are various methods of gene delivery, such as microinjection into the pronucleus and nuclear transfer (cloning), that have been already successfully applied in livestock biotechnology. At the same time, SMGT, i-GONAD and TPGD are emerging techniques that show promise in the field of gene delivery. While these techniques are still in the early stages of development, they offer exciting possibilities for the future of gene editing in both agricultural and medical applications. However, further research and testing will be required to determine their applicability and efficiency in livestock. Taking into consideration all the above-mentioned methods of gene delivery in livestock and other mammalian species, the achievement of the conditions of relatively high effectiveness and sustainable repeatability of the results might open new possibilities for the successful generation and multiplication of genome-edited nuclear recipient cells (spermatozoa, oocytes, zygotes), conceptuses and offspring using a wide variety of assisted reproductive technologies (ARTs). The latter encompass such strategies as standard *in vivo* embryo production and modern *ex vivo* embryo production with the aid of more advanced methods, including SCNT-mediated cloning and IVF by either gamete coincubation or ICSI [90,91,92,93,112,252,253,254,255,256,257,258,259].

## Figures and Tables

**Figure 1 animals-13-02580-f001:**
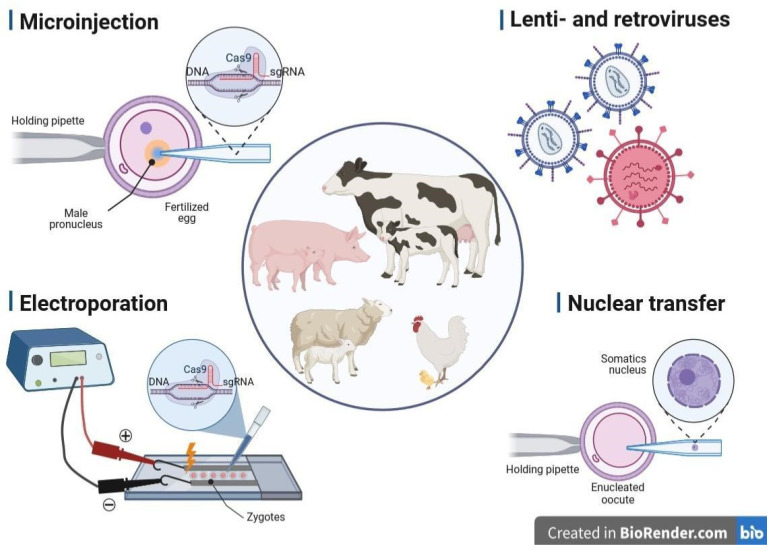
Most popular methods of gene delivery successfully applied in livestock biotechnology. This figure was created with BioRender.com (accessed on 15 June 2023).

**Figure 2 animals-13-02580-f002:**
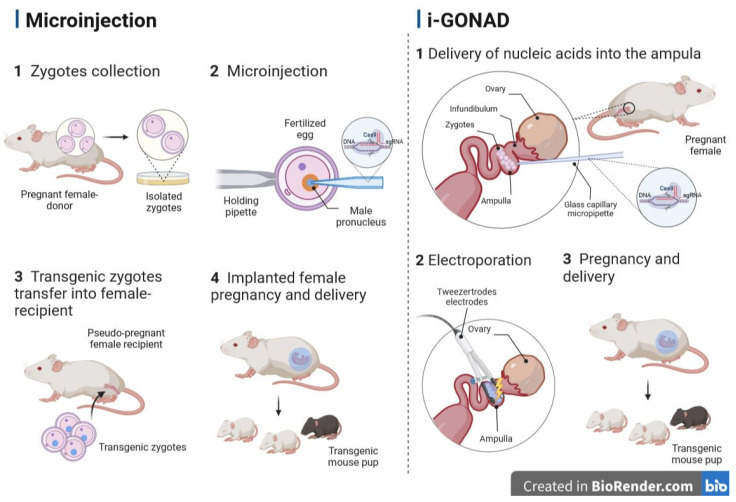
A schematic comparison between microinjection and improved genome editing via oviductal nucleic acid delivery (i-GONAD). Figure created with BioRender.com (accessed on 15 June 2023).

**Table 1 animals-13-02580-t001:** Transgenic livestock approved for industrial use.

Species	Gene	Trait	Effect	Commercial Name	References
fish	*gh1*	Production trait	fast-growing salmon: 2- to 6-fold as compared to the wild-type fish	AquAdvantage	[204]
pig	*GGTA1*	Medical use: xenotransplantation	reduces the risk of transplant rejection due to no alpha-gal sugar on cell surfaces	GalSafe	[205,206,207]
cattle	*SLICK*	Breed quality	substantially increases thermotolerance and thermoregulatory ability	PRLR-SLICK cattle	[208]
goat	*ATryn1*	Medical use: drug production	the human *ATryn1* (antithrombin-III) expressed by goats in milk	ATryn	[212]
rabbit	*C1INH*	Medical use: drug production	producing recombinant human C1 esterase inhibitor (Rhucin) in milk	Ruconest	[213,214]
chicken	*LIPA*	Medical use: drug production	express lipase A, lysosomal acid type, in eggs for long-term enzyme replacement therapy	Kanuma	[215]
fish	*mylz2*	Fluorescent protein overexpression	overexpress GFP, YFP and RFP under a strong muscle-specific *mylz2* promoter	GloFish	[217]

## Data Availability

Not applicable.

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
