# Peer review of "Perspectives in Genome-Editing Techniques for Livestock"

_animals, 2023, doi:10.3390/ani13162580_

Round 1

Reviewer 1 Report

The manuscript is nicely written and presents an informative review of gene-editing methods, tools and applications in livestock species. There are a few minor suggestions to improve different sections of the manuscript.

Choose a better list of keywords.

Line 65: change to “per se”.

Line 77: change name to “Eenennaam”

Follow the standard use of abbreviations. Full words following the abbreviations in the parentheses. For example, “genetically modified (GM)”.

Lines 119-165: I suggest providing the examples in a table with columns: species, gene, trait, effect and other details or comments, and company. Can put in an order relevant to the type of species and then the type of products, e.g., production traits, medicines etc.

Lines 166-277: Change the sequence of methods (bring 3.2 before 3.1) to suit the development history and the Fig 1.

It may also be appropriate to change the manuscript with a sequence of topics as: Tools (section 4), Methods (3) and successful livestock transgenes (2).

English language is generally used appropriately. A thorough check is suggested to fixed minor errors and, try and split the long sentences.

Author Response

We kindly thank the Reviewer #1 for his/her thorough and constructive review. We agree on all comments and tried to accomplish all required changes.

R: Line 65: change to “per se”.

A: This change has been made

R: Line 77: change name to “Eenennaam”

A: This change has been made

Follow the standard use of abbreviations. Full words following the abbreviations in the parentheses. For example, “genetically modified (GM)”.

A: This change has been made

R: Lines 119-165: I suggest providing the examples in a table with columns: species, gene, trait, effect and other details or comments, and company. Can put in an order relevant to the type of species and then the type of products, e.g., production traits, medicines etc.

A: The table has been added to the manuscript

R: Lines 166-277: Change the sequence of methods (bring 3.2 before 3.1) to suit the development history and the Fig 1.

A: This change has been made

R: It may also be appropriate to change the manuscript with a sequence of topics as: Tools (section 4), Methods (3) and successful livestock transgenes (2).

A: This change has been made

Reviewer 2 Report

17th July, 2023

Review of the Manuscript ID: animals-2520502, by J. Popova et al., entitled: “Perspectives in genome editing techniques for livestock animals” that is intended to be published as the Review paper in Animals

(separate Microsoft Word file as Reviewer Attachment for Manuscript ID animals-2520502 Animals 17th July 2023 that includes Comments to the Authors is also uploaded)

Considering research highlight, contribution of the Authors to the progress in the research area, comprehensive manner of data presentation, very well writing in English, abundance of research topics and issues presented in the form thoroughly prepared sections and subsubsections and finally diligent graphic visualization, the quality of this paper deserves praise and merits my support. The Authors have received the very high scores from me for the originality, importance of the work and the scientific value of their paper. In my opinion, the current paper provides a research highlight and insightful interpretation of biotechnological and molecular background and factors determining the application potential of a broad spectrum of  approaches used for genetic engineering and genome editing in livestock and other mammalian species. Taking all the aforementioned facts into account, I strongly recommend the Editorial Board to enable the publication of this very interesting paper in Animals after the minor revision of the manuscript will have been completed by the Authors and provided that the Authors are ready to consider all the Reviewer comments indicated below:

1) Please clarify the description of cloning by somatic cell nuclear transfer (SCNT) in the section 3. and provide more details regarding this procedure between the lines 176 and 177 of this section. To make it please re-edit the last sentence of this section and add the following sentences as below-indicated by the Reviewer:

There are other, less commonly used methods of NA delivery, such as electroporation-mediated gene transfer, viral transduction using adeno-, retro-, and lentiviral vectors, and liposomal transfection [52,53]. In turn, one of the most advanced strategies of propagating genetically engineered or genome-edited specimens in livestock species seems to be cloning by somatic cell nuclear transfer (SCNT) [54–57].  By applying SCNT-mediated cloning, genetically transformed progeny can be either created with the use of in vitro-transfected nuclear donors (somatic or stem cells) or multiplied with the use of ex vivo-expanded nuclear donor cells derived from existing gene-edited farm animals that have been formerly generated by other techniques of gene delivery [58–61]. A broad spectrum of the standard and more advanced techniques of gene delivery have been presented in Figure 1.  

2) I would like to clearly highlight that there is a lack of final paragraph (including comprehensive summary and future research directions and goals focused on assisted reproductive technologies) at the very end of the Conclusions section (between the lines 735 and 736). Therefore, missing details at the very end of Conclusions section and missing research article citations and related References are required to be added according to the Reviewer comments indicated below:

            Taking into consideration all the above-mentioned methods of gene delivery in livestock and other mammalian species, the achievement of the conditions of relatively high effectiveness and sustainable repeatability of the results might open up the new possibilities for the successful generation and multiplication of genome-edited nuclear recipient cells (spermatozoa, oocytes, zygotes), conceptuses and offspring by using a wide variety of assisted reproductive technologies (ARTs). The latter encompass such strategies as standard in vivo embryo production [67–70], and modern ex vivo embryo production with the aid of more advanced methods, including SCNT-mediated cloning [57,241,242] and in vitro fertilization (IVF) by either gamete coincubation [243–245] or intracytoplasmic sperm injection (ICSI) [246–248].

3) The following 16 References have to be added and cited in the text of manuscript (according to the re-editions required by Reviewer in the above-listed comments 1 and 2):

[54] Li, H.; Wang, G.; Hao, Z.; Zhang, G.; Qing, Y.; Liu, S.; Qing, L.; Pan, W.; Chen, L.; Liu, G.; et al. Generation of biallelic knock-out sheep via gene-editing and somatic cell nuclear transfer. Sci. Rep. 2016, 6, 33675. doi: 10.1038/srep33675.

[55] Samiec, M.; Skrzyszowska, M. Transgenic mammalian species, generated by somatic cell cloning, in biomedicine, biopharmaceutical industry and human nutrition/dietetics--recent achievements. Pol. J. Vet. Sci. 2011, 14, 317–328. doi: 10.2478/v10181-011-0050-7.

[56] Preisinger, D.; Winogrodzki, T.; Klinger, B.; Schnieke, A.; Rieblinger, B. Genome Editing in Pigs. Methods Mol. Biol. 2023, 2631, 393–417. doi: 10.1007/978-1-0716-2990-1_19.

[57] Skrzyszowska, M.; Samiec, M. Generating Cloned Goats by Somatic Cell Nuclear Transfer-Molecular Determinants and Application to Transgenics and Biomedicine. Int. J. Mol. Sci. 2021, 22, 7490. doi: 10.3390/ijms22147490.

[58] Tan, W.; Proudfoot, C.; Lillico, S.G.; Whitelaw, C.B. Gene targeting, genome editing: from Dolly to editors. Transgenic Res. 2016, 25, 273–287. doi: 10.1007/s11248-016-9932-x.

[59] Wiater, J.; Samiec, M.; Wartalski, K.; SmorÄ…g, Z.; Jura, J.; SÅ‚omski, R.; Skrzyszowska, M.; Romek, M. Characterization of Mono- and Bi-Transgenic Pig-Derived Epidermal Keratinocytes Expressing Human FUT2 and GLA Genes – In Vitro Studies. Int. J. Mol. Sci. 2021, 22, 9683. doi: 10.3390/ijms22189683.

[60] Skrzyszowska, M.; Smorag, Z.; SÅ‚omski, R.; Katska-Ksiazkiewicz, L.; Kalak, R.; Michalak, E.; Wielgus, K.; Lehmann, J.; LipiÅ„ski, D. Szalata, M.; et al. Generation of transgenic rabbits by the novel technique of chimeric somatic cell cloning. Biol. Reprod. 2006, 74, 1114–1120. doi: 10.1095/biolreprod.104.039370.

[61] Wu H, Zhou W, Liu H, Cui X, Ma W, Wu H, Li G, Wang L, Zhang J, Zhang X, Ji P, Lian Z, Liu G. Whole-genome methylation analysis reveals epigenetic variation between wild-type and nontransgenic cloned, ASMT transgenic cloned dairy goats generated by the somatic cell nuclear transfer. J. Anim. Sci. Biotechnol. 2022, 13, 145. doi: 10.1186/s40104-022-00764-6.

[241] Yamashita, M.S.; Melo, E.O. Animal Transgenesis and Cloning: Combined Development and Future Perspectives. Methods Mol. Biol. 2023, 2647, 121–149. doi: 10.1007/978-1-0716-3064-8_6.

[242] Galli, C.; Lazzari, G. 25th ANNIVERSARY OF CLONING BY SOMATIC-CELL NUCLEAR TRANSFER: Current applications of SCNT in advanced breeding and genome editing in livestock. Reproduction 2021, 162, F23–F32. doi: 10.1530/REP-21-0006.

[243] Morita, K.; Honda, A.; Asano, M. A Simple and Efficient Method for Generating KO Rats Using In Vitro Fertilized Oocytes. Methods Mol. Biol. 2023, 2637, 233–246. doi: 10.1007/978-1-0716-3016-7_18.

[244] Namula, Z.; Le, Q.A.; Wittayarat, M.; Lin, Q.; Takebayashi, K.; Hirata, M.; Do, L.T.K.; Tanihara, F.; Otoi, T. Triple gene editing in porcine embryos using electroporation alone or in combination with microinjection. Vet. World 2022, 15, 496–501. doi: 10.14202/vetworld.2022.496-501.

[245] Bevacqua, R.J.; Fernandez-Martín, R.; Savy, V.; Canel, N.G.; Gismondi, M.; Kues, W.A.; Carlson, D.F.; Fahrenkrug, S.C.; Niemann, H.; Taboga, O.A.; et al. Efficient edition of the bovine PRNP prion gene in somatic cells and IVF embryos using the CRISPR/Cas9 system. Theriogenology 2016, 86, 1886–1896.e1. doi: 10.1016/j.theriogenology.2016.06.010.

[246] Nakagawa, Y.; Kaneko, T. Rapid and efficient production of genome-edited animals by electroporation into oocytes injected with frozen or freeze-dried sperm. Cryobiology 2019, 90, 71–74. doi: 10.1016/j.cryobiol.2019.08.004.

[247] Mizushima, S.; Sasanami, T.; Ono, T.; Kuroiwa, A. Current Approaches to and the Application of Intracytoplasmic Sperm Injection (ICSI) for Avian Genome Editing. Genes 2023, 14, 757. doi: 10.3390/genes14030757.

[248] Lotti, S.N.; Polkoff, K.M.; Rubessa, M.; Wheeler, M.B. Modification of the Genome of Domestic Animals. Anim. Biotechnol. 2017, 28, 198–210. doi: 10.1080/10495398.2016.1261874.

4) There is a lack of the separate Abbreviations section in the paper. For that reason, this section should have been added to thoroughly elucidate and expand a wide range of the in-text abbreviations, which have been used by the Authors in all the subsections of their paper.

General Comment of the Reviewer:

Before the manuscript will have been accepted for publication in Animals, it requires the minor revision (according to all the recommendations of the Reviewer) and re-review to confirm the correctness of changes that will have been made by the Authors in the re-edited and resubmitted version of their paper. 

Author Response

We kindly thank the Reviewer #2 for his/her valuable comments and important addition to the bibliography. We agree with all suggestions and tried to accomplish all required changes.

1) Please clarify the description of cloning by somatic cell nuclear transfer (SCNT) in the section 3. and provide more details regarding this procedure between the lines 176 and 177 of this section. To make it please re-edit the last sentence of this section and add the following sentences as below-indicated by the Reviewer:

A: This change has been made.

2) I would like to clearly highlight that there is a lack of final paragraph (including comprehensive summary and future research directions and goals focused on assisted reproductive technologies) at the very end of the Conclusions section (between the lines 735 and 736). Therefore, missing details at the very end of Conclusions section and missing research article citations and related References are required to be added according to the Reviewer comments indicated below:

A: This change has been made.

3) The following 16 References have to be added and cited in the text of manuscript (according to the re-editions required by Reviewer in the above-listed comments 1 and 2):

A: These references have been added to the bibliography.

4) There is a lack of the separate Abbreviations section in the paper. For that reason, this section should have been added to thoroughly elucidate and expand a wide range of the in-text abbreviations, which have been used by the Authors in all the subsections of their paper.

A: A corresponding section with the requested abbreviations was added to the manuscript.